# Comparing Executive Functions in Children and Adolescents with Autism and ADHD—A Systematic Review and Meta-Analysis

**DOI:** 10.3390/children11040473

**Published:** 2024-04-15

**Authors:** Claudia Ceruti, Alessandra Mingozzi, Nicoletta Scionti, Gian Marco Marzocchi

**Affiliations:** Department of Psychology, University of Milan-Bicocca, 20126 Milan, Italy; claudia.ceruti@gmail.com (C.C.); alessandra.mingozzi97@gmail.com (A.M.);

**Keywords:** executive functions, ASD, ADHD, measures, meta-analysis

## Abstract

Two neurodevelopmental conditions, autism spectrum disorder (ASD) and attention deficit hyperactivity disorder (ADHD), have been associated with executive function (EF) impairments but the specificity of their impairments is still controversial. The present meta-analysis aimed to identify the differences in EF profiles of ASD, ADHD, and ASD+ADHD in relation to a control group of individuals with typical development (TD) and to understand whether the EF performance could change depending upon the type of measure used to assess EF (performance tests vs. questionnaires). Results from 36 eligible studies revealed that ADHD and ASD showed more difficulties than the TD group in tests and, particularly, in questionnaires. No significant differences in the EF profile emerged between ASD and ADHD when assessed through neuropsychological tests (*d* = 0.02), while significant differences emerged when assessed through questionnaires, with ADHD having higher ratings than ASD (*d* = −0.34). EF questionnaires and neuropsychological tests may catch two different constructs of EF, with the former being more predictive of everyday life EF impairments. The comparison between the double diagnosis group (ADHD+ASD) and the clinical groups pointed out that the former has a more similar EF profile to the ADHD-alone one and that it shows more difficulties than ASD-alone.

## 1. Introduction

Executive functions (EF) impairments have been fully described and linked to attention deficit hyperactivity disorder (ADHD) and autism spectrum disorder (ASD) [1]. Although no unanimous agreement has been reached on the definition of EF, most definitions describe them as a broad family of mental top-down processes used during the selection and execution of goal-directed thoughts and behaviors [2]; they are essential for cognitive, social, and psychological development.

ADHD is a neurodevelopmental condition primarily characterized by symptoms in two behavioral domains: inattention and/or hyperactivity and impulsivity [3]. Many studies have observed EF impairments in ADHD [4,5] but a controversy over whether executive dysfunction should be identified as the core impairment or just as a secondary feature of the condition remains [6]. If we consider EF impairment as one of the most important characteristics in children with ADHD, then we will expect to observe lifelong difficulties in this domain across adulthood. However, empirical findings in this field are yet not clear. Given the heterogeneity of the results, it does not seem helpful to think of a unitary model where executive dysfunction represents the single-core impairment in ADHD. Instead, the conceptualization of this disorder as a neurodevelopmental condition with heterogeneous origins and manifestations seems more appropriate [7]; this heterogeneity becomes evident considering the impairments in other domains such as delay aversion, i.e., the tendency to avoid delay and obtain an immediate reward. For this reason, it can be assumed that ADHD includes problems at the inhibition level and problems in motivational and high-affective processes.

Unlike ADHD, executive impairment has not generally been conceptualized as one of the distinctive features of ASD, a neurodevelopmental condition described by a dyad of symptoms that includes social communication inadequacy combined with patterns of restricted/repetitive behaviors or interests [3]. ASD has been described as primarily characterized by social cognition impairments; however, it should be noted that many people with ASD often exhibit an impairment in the EF domain [8,9], although different studies have reached mixed conclusions on the type of affected function. It is known that social cognition is related to EF in ASD: as Geurts et al. (2010) pointed out, cognitive flexibility requires the ability to shift between tasks and strategies, in the same way as a solid Theory of Mind (ToM) requires the ability to change the point of view, being able to take and understand someone else’s perspective [10].

Regarding the co-occurrence of these two disorders, while the DSM-IV did not allow a comorbid diagnosis of ADHD and ASD [11], as they were mutually exclusive, the fifth edition of DSM (DSM-5) [3] admits the possibility of a co-occurrence between these two conditions. The EF profile of ADHD+ASD, however, is not yet completely clear; in fact, some studies have described important differences in attentional functions between ASD and ADHD individuals, while others suggest that these disorders show identical impairments [12]. A research question indeed concerns their EF profile, questioning what kind of EF impairments are typically associated with the comorbidity of ADHD and ASD. This question, however, does not have an easy answer, as ADHD and ASD are often characterized by similar EF weaknesses; Carter Leno and colleagues (2018) have shown the limit of using EF profile to discriminate between ADHD and ASD, since in their study, they failed to differentiate these disorders using EF tasks and found many EF impairments shared by the ADHD group and the ASD group [13].

Since EF is a multidimensional construct, there is still no unanimous agreement on which specific functions are part of this umbrella term. However, according to a review of more than 100 EF studies [14], three main core EF studies were the most frequently mentioned across studies, in line with Miyake’s model [15]: inhibition, working memory (WM), and cognitive flexibility. Inhibition is the capacity to “control one’s attention, behavior, thoughts, and emotion to over-ride a strong internal predisposition or external lure, and instead do what is more appropriate or needed” [2] (p. 137); inhibition can be captured by a task that requires the subject to inhibit either a thought or an action, as a Stroop task and Go/No-Go task. WM can be defined as the ability to hold in mind and mentally manipulate information simultaneously, captured by a test that required an active manipulation of visuospatial or verbal material such as the Corsi block or backward digit span. At last, cognitive flexibility, also defined as set-shifting, refers to various abilities, including changing perspectives both spatially and interpersonally and adapting to new and changed demands without clinging to a rigid perspective. The cognitive flexibility test requires a shifting or a flexible adjustment to new rules, e.g., the Wisconsin Card Sorting Test, Trail Making test). Other functions, higher-order EF, such as planning, reasoning, and problem solving, develop from these core processes [16]. Planning tests require children to program the right action with mental anticipation (e.g., Tower test), while problem-solving tasks challenge thinking and assess fluid intelligence (i.e., Raven matrices and Block Design of WISC-IV).

Historically, EF has been conceptualized as pure logic and cognitive processes but recently, the role of motivational aspects has emerged [17]. For this reason, sometimes EF can be divided into two subgroups called “hot” and “cool” EF. This difference concerns the type of cognitive process involved as well as the neural networks; cool EF concerns the strictly cognitive level and is associated with the dorsolateral prefrontal cortex, whereas hot EF concerns the emotional/motivational level in situations where the psychological process is driven by emotions exerting their interference and include the orbital and medial prefrontal cortex [18]. Hot EF is the type of EF needed during self-regulation, a process based on effortful control, meaning the ability to “maintain optimal levels of emotional, motivational and cognitive arousal” [2,19] (p. 152).

Methods used to evaluate EF can be divided into two categories that do not provide the same information: questionnaires and neuropsychological tests. In order to obtain a detailed assessment, both measures (questionnaires and neuropsychological tests) should be used as they often do not reflect the same constructs or catch the same information [20]. Performance-based tests are well structured and generally provide an assessment under optimal conditions; therefore, individuals with problem behaviors may sometimes perform adequately on this type of test. On the other hand, rating measures do not assess optimal performances: they focus on everyday life contexts, providing a more ecological assessment although influenced by potential biases related to the context and to the informant.

Previous studies have investigated EF separately in ADHD or ASD, comparing them to a typical developing (TD) group, leading to the homogeneous conclusion that both clinical conditions are usually impaired in this domain, showing more EF difficulties than TD, although findings remain inconclusive regarding the severity of the impairment and the specificity of the impaired EF-component. Considering the comparison between ADHD and ASD, several studies have also compared these two disorders alone, leading to heterogeneous conclusions: while some of them found that the two groups were very similar in their EF profiles [21], other studies [22] proposed that ADHD is mainly associated with an inhibition weakness (as proposed by Barkley in 1997 [23]), whereas in ASD the EF components that have been found to be more impaired are cognitive flexibility and planning [24,25,26]. A recent systematic review and meta-analysis by Townes and colleagues (2023) evaluated EF differences between individuals under 19 years of age with a diagnosis of ASD and ADHD, examining 58 articles [27]. Townes et al. found no differences in EF profiles between individuals diagnosed with ASD and ADHD, stating that although they are two distinct clinical disorders, they share some underlying mechanisms. This meta-analysis confirmed that both ADHD and ASD showed EF impairments compared to the TD group. No distinction between the types of assessment (i.e., neuropsychological tests vs. questionnaires) was taken into account.

However, most of the studies have reached the same conclusion: an EF weakness characterizes both ADHD and ASD and the difference between them does not lie in the global EF (impaired in both disorders) but possibly within the single EF components that may be more or less impaired according to the clinical condition.

The current meta-analysis was undertaken to clarify the different EF profiles of ADHD and ASD, using a control group formed by individuals with typical development. Our aim was to shed more light on a topic with contrasting and non-univocal results; clarifying if the EF profile of these two disorders could help researchers and clinicians to understand whether EF difficulties may be considered a specific marker associated with one disorder more than the other. Moreover, we were interested in finding if the EF profile may vary based on the type of instrument used to assess it (i.e., neuropsychological test vs. questionnaires) since the use of different instruments in clinical evaluation should follow a reasoned criterion such as, among others, choose the most appropriate instrument in order to best capture and describe the clinical characteristics of someone’s individual functioning.

Based on the literature, the following research questions were taken into account:(1)Firstly, we examined the EF performance of the two clinical groups (ADHD and ASD) compared to the TD group;(2)Secondly, we aimed to search for possible differences within the single EF components by comparing the ADHD group and the ASD group;(3)Thirdly, we aimed to understand whether differences in the overall EF performance changed depending on the type of assessment measures: performance-based neuropsychological tests to assess the efficiency of high-order cognitive processes vs. questionnaires to assess success in goal pursuit; to address this question, we carried out two distinct analyses, since, as mentioned above, these instruments do not assess the same construct. We aimed to obtain a wider comprehension regarding the differences in EF between individuals with ADHD and ASD based on these two measures;(4)Furthermore, an additional goal was to investigate the EF profile of the double-diagnosis group (ADHD+ASD). We included the group with comorbidity in order to understand if a double diagnosis would be associated with higher, similar, or lower executive impairment compared to the single diagnosis (ADHD-only and ASD-only).

## 2. Methods

### 2.1. Inclusion and Exclusion Criteria

The eligible studies had to meet the following inclusion criteria: (a) studies with comparisons of children/adolescents with ADHD, children/adolescents with ASD and TD children/adolescents as the control group; (b) participants with IQ ≥ 70; (c) at least one measure related to EF; (d) at least one measure of EF had to be a direct measure (experimenter-administered or computer-based neuropsychological test) or an indirect measure (questionnaire); (e) at least ten participants per group, as smaller samples could increase the probability of publication bias; (f) diagnosis of ADHD, ASD, or double diagnosis made by appropriate professional figures referring to the main diagnostic classification systems (DSM IV, IV-TR, 5, and International Classification of Diseases, 10 [ICD 10]). The use of different versions of DSM was because many studies were conducted before the release of the current version of DSM and they included participants with different clinical diagnoses that later converged in the “ASD” diagnostic category of the current version of DSM, DSM-5; (g) paper written in English; and (h) studies from 1999 to 2021 were considered.

We excluded those studies whose participants were outside the age range we wanted to assess (6–18 years). We decided to consider this age range for two reasons: primarily, this was because the diagnosis of ADHD under the age of 6 can be unreliable since the hyperactive and impulsive symptoms under the age of 4 are hardly differentiated from the hyperactive manifestations that can be typically observed during early infancy and the inattention symptoms become predominant during the primary school years [3]; secondly, because over the age of 18, the clinical manifestations of EF and the instruments used to assess them may be different and the EF development usually reaches a plateau during the young adulthood (around 22–23 years old). Then, we excluded all the studies evaluating EF through neuro-functional methods such as fMRI, MRI, EEG, or PET; however, we included studies containing an assessment of EF associated with neuro-functional methods, if they reported data of accuracy/reaction times of behavioral tasks. The outcomes we assessed were presented in the form of accuracy (or error rates) or reaction time ([RT] or speed). In addition to the three groups described above, we included those studies containing a double-diagnosis group. For a detailed description of each study, please see Appendix A.

### 2.2. Data Identification and Screening

In accordance with the PRISMA statement [28], a systematic search strategy was implemented to retrieve eligible studies to include in the meta-analysis. The research plan has been preregistered on the Prospero website (https://www.crd.york.ac.uk/prospero/display_record.php?ID=CRD42021232689).

We searched through the following databases: PubMed, PsychInfo, Web of Science, ProQuest Dissertations and Theses Global, e-thesis online service (EThOS), and DART-Europe E-theses Portal to retrieve all the potential published and unpublished studies (such as doctoral dissertations) representing the “gray literature”. In addition to this, we searched the reference lists of retrieved articles to find further potentially admissible studies. Our search strategy consisted of a combination of the terms “executive functions”, “ADHD”, “ASD”, and their synonyms (see Appendix B for the entire search string) implemented through the different databases. After excluding duplicates, 5425 records were screened independently by the second author and the third author based on inclusion/exclusion criteria. The agreement rate in this phase was 99%. Of these articles, only 102 were assessed for eligibility and screened full-text by the second and the third authors. Four studies were selected from the reference list; therefore, in total, 106 full-text articles were screened. The agreement rate in this phase was 91%; any disagreement about the eligibility of a study was solved through discussion with the fourth author. Finally, we were able to identify 36 articles that met all the inclusion criteria and were included in the meta-analysis, with a total of 505 effect sizes (437 studies including neuropsychological tests and 68 studies including questionnaires). The flow chart presenting the method of literature search and exclusion criteria is shown in Figure 1.

During the coding phase, the second and the third authors coded different characteristics of the sample following a predefined coding schema. This included bibliographic information (title, authors, country of publication, year of publication, type of publication [published vs. unpublished]), information about participants (sample size, mean age, mean IQ and number of females in each group, presence vs. absence of the double diagnosis group, presence vs. absence of washout—this term usually refers to a suspension from medication that could be discontinued 24 to 48 h before the testing—from any medication such as stimulant, anti-depressants, anti-anxiety, or anti-psychotic drugs), and information about the instruments used to assess EF (type of measure, type of EF domain assessed, type of task, type of outcome measure [accuracy vs. errors], and use of standardized test). EFs were coded following Diamond’s hierarchical model [2], containing three core EFs (inhibition, working memory, and cognitive flexibility) and two high-order EFs (planning and problem solving). We considered “neuropsychological tests” those instruments that directly assessed EF, based on a performance that was later interpreted based on normative standards (indeed, they are also called “performance-based tests”). We classified neuropsychological tests as measures of different EF based on the description mentioned in the introduction. With regard to the questionnaires, we considered those instruments that require a person (parents/teachers/children themselves) to evaluate EF in everyday life; three main questionnaires converged within this type of instrument: BRIEF (first edition) [29], early adolescent temperament questionnaire-revised [30], and attentional control scale [31]. Information on the reliability of the measures used in the collected studies is provided in Appendix A. Details of the individual studies are presented in Appendix A. The database collected for the purpose of this meta-analysis can be retrieved online at Appendix A Data.

### 2.3. Data Extraction

First, duplicates have been searched and removed by software (Excel 2020). Second, the title and abstract of the rest were screened by the second author according to inclusion and exclusion criteria. For articles without an available abstract or lack of information for a decision, the full-text version has been reviewed. All retrieved articles have been screened independently by the third author based on titles and abstracts. Studies that did not meet the defined criteria had been discarded. Furthermore, the reference lists of former reviews and relevant studies have been scanned by the second author to pick up potentially eligible studies not captured by the literature search.

Then, the second and the third authors independently reviewed the full text of the identified articles to identify studies for final inclusion. In doing so, the key information has been extracted and kept in Excel sheets. Disagreements were resolved through discussion and with a third review team member, if necessary. Extracted data include bibliographical information, sample demographics (e.g., age, sex), study setting, study design and methodology, information about the included measurement scales, results relating to the research question, and information relevant to quality assessment.

#### Population

All the participants from the 36 papers included in the meta-analysis were children and adolescents in the age range of 6 to 18 years old, with an IQ ≥ 70, and were members of four possible groups: ADHD, ASD, ADHD+ASD, and TD. The ADHD group included subjects with predominantly inattentive type, predominantly hyperactive type, and combined type; the ASD group included participants with a diagnosis of “Autistic Disorder” but also of “Asperger Syndrome” or “Pervasive developmental disorder not otherwise specified, PDD-NOS” as the DSM-IV-TR described [11]. The comorbidity group included individuals with symptoms related to both diagnostic categories and who met the criteria for both diagnoses [3]. Finally, the TD group included children and adolescents without any psychiatric history or clinical diagnosis and with a normal history of development. Subjects from the clinical groups were considered free from the influence of any medication if the authors specified that they selected only subjects without any pharmacological treatments or if they specified that they had an adequate period (24 to 48 h) of washout in order to eliminate from their body the medication. Instead, they were considered under the influence of medication if they did not have an adequate washout period or if they had it only for stimulant medication and not for other drugs such as anti-depressants and anxiolytics. However, within the included studies, only four studies reported just a partial washout and six studies gave no information about medication use.

### 2.4. Meta-Analytic Approach

The statistical software JASP version 0.14.1 (JASP team, 2020) was used to perform the meta-analysis comparing group effect sizes. A random-effects model was used for all the analyses due to population heterogeneity [32] and the maximum likelihood method was used to estimate the parameters. We calculated a standardized effect size (Cohen’s *d*) for each study: the effect sizes had been computed for groups with different sample sizes by adjusting the pooled standard deviation calculation with weights for the sample sizes. This approach is overall identical to Cohen’s *d* (Cohen’s *d* = *M*1 − *M*2/*SD* pooled) with a correction of a positive bias in the pooled standard deviation. Standard group differences are classified as small (*d* = 0.20), moderate (*d* = 0.50), and large (*d* = 0.80; Cohen, 1988). Additionally, a confidence interval for the effect size and the desired confidence coefficient was computed according to Hedges and Olkin (1985) [32]. For each study, we calculated an effect size for the mean difference in EF components (i.e., inhibition, verbal, and visuospatial working memory, cognitive flexibility, planning, and problem solving) between the groups compared in the meta-analysis. Thus, each study might contribute to the effect size for multiple EF components, violating the principle of independence [33]. Since we were interested in knowing if EF varied across different EF domains between children diagnosed with ASD and ADHD, we decided to include multiple EF outcomes within the studies. This would allow for investigation of whether ASD and ADHD EF differ by EF domain. The heterogeneity of the effects was assessed with *Q* statistics [34]. As declared in pre-registration, in addition to the principal analysis, we have conducted moderation analyses of continuous moderators through a meta-regression analysis and categorical moderators through *Q* statistics to investigate potential sources of heterogeneity. These analyses can be found in Appendix A. To address our research questions, we divided the analysis into five contrasts:

The first three contrasts investigated the differences between ADHD, ASD, and TD (contrast 1: ADHD vs. ASD; contrast 2: TD vs. ADHD; and contrast 3: TD vs. ASD);

The fourth and the fifth contrasts addressed the additional research question about the EF profile of the double diagnosis group (contrast 4: ADHD vs. ADHD+ASD and contrast 5: ASD vs. ADHD+ASD).

We converted the data so that in contrast 1, a positive effect size represented a finer performance in ADHD, in contrast 2 and 3 a positive effect size represented a finer performance in TD, and in contrast 4 and 5 a positive effect size indicated, respectively, that ADHD and ASD had a higher performance than the double diagnosis group. Only in the first contrast (ADHD vs. ASD) did we investigate if the global EF performance varied across the EF component, using the “EF component” as a moderator. To address the research question about the type of instruments used to assess EF, for each contrast, we first investigated the potential influence of the variable “type of measure”. If it showed a significant effect, we proceeded with separate analyses between direct measures (neuropsychological tests) and indirect measures (questionnaires). Therefore, for each of the three contrasts, we separated the data by the type of EF instrument used (neuropsychological tests or questionnaires) and ran analyses separately.

### 2.5. Study Quality and Risk of Bias

The included studies have not been assessed for quality because extracted data from eligible studies are cross-sectional in nature. However, the findings have been examined for publication bias using a Funnel plot and by comparison of the results from studies from peer-reviewed journals and the grey literature.

### 2.6. Publication Bias

Given that published studies, compared with unpublished studies, are more likely to report significant findings, meta-analyses can potentially overestimate effect sizes [35]. To ensure the present meta-analysis was not influenced by publication bias, we explored the funnel plots (see online resource four for the funnel plots of the five contrasts) and checked the rank correlation (Kendall’s tau). Contrast 1 had a balanced funnel plot and a non-significant Kendall’s tau *p*-value (Kendall’s tau = 0.029, *p* = 0.615), while contrast 2 and 3 showed a less balanced distribution and a significant Kendall’s tau *p*-value (*p* < 0.001 and *p* = 0.002), meaning possible asymmetry and publication bias. However, several other studies confirmed significant differences in the EF profile between TD and the two clinical groups, with TD having significantly higher EF performance than ADHD and ASD; therefore, the significant *p*-values were not a major concern. Funnel plots of the comorbidity group were balanced and showed no indication of publication bias (Kendall’s tau_ADHD vs. ADHD+ASD_ = 0.183, *p* = 0.094; Kendall’s tau_ASD vs. ADHD+ASD_ = 0.124, *p* = 0.258).

## 3. Results

### 3.1. Descriptive Statistics

Of the 36 articles included in the meta-analysis, 33 were published papers and 3 were doctoral dissertations. Table 1 shows the descriptive characteristics of the sample included in the meta-analysis. In total, 4760 participants across groups were included in the meta-analyses. Results show that the samples did not significantly differ in age (F(3, 115) = 0.988, *p* = 0.401) or in the number of females (F(3, 109) = 1.80, *p* = 0.151), whereas there was a significant difference in IQ between the groups (F(3, 106) = 11.6, *p* < 0.001), with the post hoc test (Tukey HSD) indicating that the TD group had higher IQ than ADHD (*p* < 0.001), ASD (*p* < 0.001), and ADHD+ASD (*p* = 0.011), whereas the clinical groups did not differ from each other.

### 3.2. Main Analyses: TD vs. ADHD Comparison

A significant effect was found comparing TD to ADHD, Z = 10.471, *p* < 0.001, *d* = 0.73 (SE = 0.07, CI = [0.594, 0.867], *Q*(140) = 946.833, *p* < 0.001. The type of measure resulted in a significant value (*Q* = 53.658, *p* < 0.001); therefore, we separated the analyses.

Neuropsychological tests. ADHD had significantly lower performances than TD, *Z* = 9.338, *p* < 0.001, *d* = 0.54 (SE = 0.058, 95% CI = [0.428, 0.654]) meaning that TD performed more effectively than ADHD with a medium effect size. Significant heterogeneity was found across studies, *Q*(120) = 493.605, *p* < 0.001.

Questionnaires. The analysis revealed that ADHD had significantly lower performances than TD even in questionnaires, *Z* = 9.189, *p* < 0.001, with a Cohen’s *d* = 1.76 (SE = 0.192, 95% CI = [1.387, 2.139] and the effect is considered of very large magnitude. Heterogeneity was still significant, *Q*(19) = 176.523, *p* < 0.001.

### 3.3. Main Analyses: TD vs. ASD Comparison

TD compared to ASD resulted in a significant effect, *Z* = 11.557, *p* < 0.001, *d* = 0.64 (SE = 0.055, CI = [0.532, 0.749]); *Q*(140)= 659.172, *p* < 0.001. The type of measure resulted in a significant value (*Q* = 35.984, *p* < 0.001), so we separated the analyses.

Neuropsychological tests. ASD had significantly lower performances than the TD group, *Z* = 10.333, *p* < 0.001, with a Cohen’s *d* = 0.51 (SE = 0.049, CI = [0.412,0.605], meaning that the control group showed higher EF performance than ASD, with a medium effect size. Heterogeneity was found across studies, *Q*(120) = 391.367, *p* < 0.001

Questionnaires. ASD had a higher rating than TD in questionnaires, *Z* = 8.589, *p* < 0.001. Cohen’s *d* = 1.34 (SE = 0.156, CI = [1.036, 1.649]), which meant that TD outperformed the ASD group with an effect of large magnitude. Significant heterogeneity was found, *Q*(19) = 122.630, *p* < 0.001.

### 3.4. Main Analyses: ADHD vs. ASD Comparison

No significant effect was found comparing ADHD to ASD on general EF performance, *Z* = −0.577, *p* = 0.565, Cohen’s *d* = −0.03 (SE = 0.058; 95% CI [−0.15, 0.08]). Given the high heterogeneity (*Q*(140) = 686.535, *p* < 0.001), we proceeded to test if the “type of measure” (i.e., neuropsychological tests vs. questionnaires) could moderate the magnitude of the EF difference between groups. The impact of the “type of measure” resulted in a significant value (*Q* = 4.796, *p* = 0.029), so we decided to report the difference in EF based on the type of instruments used for each contrast.

Neuropsychological tests. No significant effect was found on EF performance assessed by direct tasks, *Z* = 0.293, *p* = 0.769; Cohen’s *d* = 0.02 (SE = 0.064, 95% CI = [−0.11, 0.14]). There was heterogeneity between studies, *Q*(129) = 584.756, *p* < 0.001. Since one of the research questions was to understand if EF performance could change between groups based on the EF domain, we used the “EF component” as a moderator to investigate potential differences. No significant differences emerged on the EF domain, *Q* = 7.203, *p* = 0.206, with none of the single EF domains (i.e., inhibition, flexibility, visuospatial WM, verbal WM, planning, and problem solving) reaching a significant value (*d* = −0.129; *d* = 0.164; *d* = 0.183; *d* = −0.48; d = 0.18; and *d* = 0.268). This analysis, however, contributed to partially decreasing the heterogeneity (*ΔQ* = 2%).

Questionnaires. On the contrary, questionnaires resulted in a significant effect because the magnitude of the EF difference varied between ADHD and ASD, *Z* = −2.684, *p* = 0.007. *d* = −0.34 (SE = 0.125, CI = [−0.58, −0.09]), meaning that ADHD was rated higher (more EF impairment) than ASD in questionnaires, with a low magnitude effect size. The test for heterogeneity was significant, *Q*(19) = 78.097, *p* < 0.001.

Table 2 reports the results of the EF differences in the first three contrasts globally and divided by the type of measure.

For a detailed description of the effect sizes for each study, please see Table 3, which shows the results of the standardized differences between ADHD vs. ASD, TD vs. ADHD, and TD vs. ASD.

### 3.5. Main Analyses: ADHD vs. ADHD+ASD Comparison

Comparing ADHD to the double diagnosis group resulted in a non-significant effect, *Z* = 0.517, *p* = 0.605, associated with a low Cohen’s *d* = 0.04 (SE = 0.08, CI = [−0.12, 0.20]). This result meant that on the EF profile assessed through neuropsychological tests, the ADHD group and the double diagnosis group did not differ. The test for the residual heterogeneity was still significant, *Q*(36) = 94.507, *p* < 0.001

### 3.6. Main Analyses: ASD vs. ADHD+ASD Comparison

On the contrary, comparing ASD to ADHD+ASD resulted in a significant effect, *Z* = 3.648, *p* < 0.001, *d* = 0.22 (SE = 0.06, CI = [0.101, 0.335]), meaning that, in neuropsychological tests, the ASD group generally performed more effectively than the double diagnosis group, with an effect size of small magnitude. The heterogeneity statistic was significant, *Q*(36) = 53.456, *p* = 0.031.

### 3.7. Moderation Analyses

Moderation analyses have been carried out to assess where the different variables could affect the comparison between groups. IQ difference, age difference, and sex distribution difference were the continuous variables taken into account for what concerned neuropsychological test results; IQ difference was a significant moderator between ADHD and ASD (*Z* = 3.445, *p* < 0.001) and age difference between TD and both ADHD (*Z* = 3.915, *p* < 0.001) and ASD (*Z* = 3.757, *p* < 0.001). The medication (washout vs. non-washout) and type of publication (published vs. dissertation), our categorical moderators for the neuropsychological test results, did not show any significant effect.

The same continuous variables were considered in assessing the questionnaire results: IQ difference was a significant moderator between TD and ADHD (*Z* = 11.44, *p* < 0.001) and sex distribution difference again between TD and ADHD (*Z* = −2.129, *p* = 0.033). Medication (washout vs. non-washout) and rater (parent vs. teacher vs. children) were the two categorical moderators taken into account for the questionnaire results. Rater moderated the strength of the EF differences in all three contrasts: ADHD vs. ASD (*Q*(19) = 21.059, *p* < 0.001), TD vs. ADHD (*Q*(19) = 26.309, *p* < 0.001), and TD vs. ASD (*Q*(19) = 17.111, *p* < 0.001).

Information about moderation analyses can be retrieved in the Appendix A, containing all the categorical and continuous moderators assessed for each contrast.

## 4. Discussion

The current meta-analysis was undertaken with the aim of comparing the EF profiles of two clinical groups, ADHD and ASD, in relation to a control group of individuals with typical development and to understand whether the EF performance could change the type of measure used to assess EF (performance tests vs. questionnaires). To address these purposes, we collected data from different studies that had to comply with specific selection criteria, leading us to select 36 final studies included in the meta-analysis.

Our first aim was to compare the overall EF profile between two clinical groups (ADHD and ASD) and a control group of TD children. We expected an overall finer EF performance in the TD group compared to the two clinical groups. As we hypothesized, ADHD and ASD both had impaired EF profiles compared to the TD group, with effect sizes of large magnitude (*d* = 0.73 and *d* = 0.64). This finding is consistent with the results obtained in previous studies, i.e., ADHD and ASD are clinical conditions characterized by a weakness in the EF domain [8]. The direct comparison between ADHD and ASD led to interesting findings: ADHD and ASD showed minimal differences in their EF when they were assessed through neuropsychological tests and questionnaires together (*d* = −0.03). The same conclusion was drawn by Townes and colleagues (2023): as shown by neuropsychological tests and questionnaires, children and adolescents with ASD and ADHD have similar EF profiles [27].

Secondly, we aimed to search for possible differences within the single EF components by comparing the ADHD group and the ASD group (according to the literature, ASD would be associated with a higher impairment in the flexibility domain whereas ADHD would be associated with a higher impairment in the inhibition domain). In order to investigate this research question, we first needed to understand whether the EF profile of these disorders would vary according to the instruments used to assess it since the differentiation within the single EF components is more appropriate considering the neuropsychological tests rather than questionnaires; therefore, we used “type of instrument” as a moderator to investigate its influence: a partly different perspective emerged when we divided the analyses according to the type of instruments used to evaluate EF, since this moderator showed a significant *p*-value in the first three contrasts, meaning that the differences in EF varied based on the type of instruments used.

This finding is connected to the third aim of the study: understanding whether differences in the overall EF performance changed depending on the type of assessment measures, i.e., performance-based neuropsychological tests to assess the efficiency of high-order cognitive processes vs. questionnaires to assess success in goal pursuit; to address this question, we carried out two distinct analyses, since, as mentioned above, these instruments do not assess the same construct. According to the literature, neuropsychological tests and questionnaires assessed different domains and therefore were associated with different results; particularly, questionnaires were overall associated with a bigger effect size, reflecting a greater impairment in the EF of ADHD children (compared both to TD and ASD children). Considering the results of the analyses that we ran, the comparison between the TD group and the two clinical groups confirmed the trend previously shown: ADHD and ASD showed lower performance than TD both in neuropsychological tests and in questionnaires, although the magnitude of the EF difference was much bigger when the instruments used were questionnaires (*d*_(TD vs. ADHD)_ = 1.76, *d*_(TD vs. ASD)_ = 1.34) compared to the tests (*d*_(TD vs. ADHD)_ = 0.54, *d*_(TD vs. ASD)_ = 0.51). However, the most interesting finding emerged when the EF performance of ADHD and ASD was compared separately for neuropsychological tests and questionnaires.

When the EF of ADHD and ASD were assessed through neuropsychological tests, our results showed no major differences between them (*d* = 0.02), confirming that their EF profiles are very similar when assessed through performance-based measures such as neuropsychological tests. This finding was in line with what Rommelse and colleagues (2011) stated in their review about endophenotypes common in Autism and ADHD [9]. They concluded that individuals with ADHD and ASD “are more alike in their EF difficulties than they are different” [9] (p. 1377). Our results support this finding. Both the global EF performance and the singular EF domains (inhibition, flexibility, visuospatial WM, verbal WM, planning, and problem solving) did not reach a significant *p*-value. Their effect sizes were all equally small in magnitude, except for the domain of verbal WM (addressed only by 2 of the 36 studies considered). The effect size related to this domain was of medium magnitude but it still did not reach a significant *p*-value (*d* = −0.129; *d* = 0.164; *d* = 0.183; *d* = −0.48 *d* = 0.18; and *d* = 0.268). We decided to investigate the moderator regarding the specific EF domain only in neuropsychological tests, as the categorization of EF in different sub-components seems to adapt better to these kinds of instruments rather than questionnaires. Neuropsychological tests, indeed, tap cognitive functioning in a specific setting, while questionnaires capture important, ecologically valid behavioral variations in everyday life. The lack of differentiation within EF sub-components between the two disorders clarifies our second research aim and disconfirms our hypothesis that ADHD would have shown a greater impairment in the inhibition domain while ASD would have shown a greater difficulty in the flexibility domain. This analysis, in fact, showed no endophenotypic differences within the single EF-domains between ADHD and ASD. Thus, the result from our meta-analysis does not confirm research lines that see the two disorders as dissociable, with ASD being more associated with impairments in planning and flexibility while ADHD is more compromised in the inhibition and verbal WM domains [26]. The construct of “dissociation”, however, is challenging to apply to the field of developmental neuropsychology because, in a developing brain, it is hard to find wholly preserved or completely impaired domains. Indeed, the cognitive specialization of the brain into separate subsystems seems to emerge later [67].

The lack of difference in EF between ASD and ADHD in our meta-analysis was limited to the neuropsychological tests. In fact, when their EF was assessed through questionnaires, the comparison showed that ADHD obtained higher ratings than ASD with an effect size of small magnitude (*d* = −0.34). Therefore, our meta-analysis shows that, when assessed through questionnaires, ADHD appears to be characterized by a greater executive impairment than ASD, a weakness that would be captured only by those instruments that assess daily life situations. These results help us to clarify the research question regarding the differences between neuropsychological tests and questionnaires. It made clear that the EF differences between the groups were heightened when the instruments used to assess them were questionnaires. Regardless of the contrast, the effect sizes were much bigger when the measures were questionnaires instead of neuropsychological tests.

The difference between EF tests and questionnaires is not a novelty: Barkley and Murphy (2010) [68] proved that neuropsychological tests of EF were not very predictive of the occupational impairment experienced every day by people with ADHD, as they showed weak relationships with measures of occupational impairment, while EF-questionnaires were strongly predictive of the occupational-impairment measures mentioned above, managing to capture daily life activities and daily life difficulties experienced by people with ADHD. It can be assumed that indirect measures of EF (e.g., reports, scales, and questionnaires) consist of more ecological tasks, based on more naturalistic settings and therefore on more representative environmental conditions. This could be the reason they seem more able to capture differences in the EF performance in daily life, whereas neuropsychological EF tests could fail to discriminate between neurodevelopmental disorders (in this case ADHD and ASD), as they show a weak correlation to daily life difficulties. This concept is consistent with the results of Gardiner and colleagues’ study (2017) [69], where they evaluated EF of ASD and TD using both neuropsychological tests and questionnaires. They found no difference in EF using neuropsychological tests, whereas using a parent-form questionnaire, they found a significant difference between the two groups, with ASD being associated with higher ratings and greater executive impairment in daily life. They conclude that this difference was due to the fact that questionnaires assess someone’s ability to “manage their day-to-day EF-related behavior” [69] (p. 1). Instead, neuropsychological tests assess children’s performance under specific and optimal circumstances, requiring the children to do their best. Other studies before had found little or no correlation between EF questionnaires (mostly BRIEF) and EF tests, leading to the conclusion that EF questionnaires and neuropsychological tests may catch two different constructs of EF [20]. Doebel (2020) also distinguishes between neuropsychological tests and questionnaires but states that performance-based evaluations and rating scales may not assess two different constructs; they rather examine a behavioral control exercised in different ways and in different environments that activate different types of beliefs and values [70]. It can be concluded that, although both tests and questionnaires are important and valid measures of EF, the results suggest that the questionnaires seem more capable of capturing daily life impairment in the EF domain. Therefore, this can be useful to discriminate the functioning between different clinical conditions.

Finally, an additional goal was to investigate the EF profile of the double-diagnosis group (ADHD+ASD). Given the small number of studies in the current literature that investigate the EF profile of the double-diagnosis group, we tried to clarify the performance of ADHD+ASD compared to the single-diagnosis groups. This, however, could be performed only for the neuropsychological tests, as we found only one study focusing on the EF performance of ADHD+ASD through questionnaires. The comparison between the double diagnosis group and the clinical groups pointed out that, when assessed through neuropsychological tests, the EF profile of ADHD+ASD is more similar to the EF profile of ADHD-alone while it shows a poorer performance than ASD-alone. This finding supports other research lines [9,26], stating that, considering the EF domain, ADHD and ADHD+ASD share several commonalities, not only on the behavioral level but also on the neuro-functional level. For instance, the double-diagnosis group shows abnormalities in the basal ganglia, present in ADHD alone but not in ASD alone [71]. As stated above, however, too few studies have focused on the comparison of EF between the double-diagnosis group and the single-diagnosis groups (especially ASD-alone), in fact, we were able to find only 11 articles that included a group with comorbid ADHD and ASD; therefore, further investigation and studies are needed in order to draw stronger conclusions.

Given the high and significant heterogeneity we found across all contrasts, we carried out moderation analyses, trying to understand the influences of other potential factors. Considering neuropsychological tests, the exploration of moderators pointed out that IQ difference was a significant moderator between ADHD and ASD, whereas age difference moderated the strength of the EF difference between TD and both ADHD and ASD. The first finding led us to recognize that when the IQ difference between ADHD and ASD was wider (i.e., when ADHD had a higher IQ than ASD), the difference in EF also increased between the two groups. As we said before, it is known that the construct of IQ is related to the construct of EF and previous research has hypothesized that an average or a high IQ could attenuate EF difficulties in clinical samples [23,72]. It is also known that, contrary to ASD samples, ADHD samples are not generally associated with low IQs [73]. Therefore, it is possible that, although in our meta-analysis no significant IQ differences emerged between ADHD and ASD, ADHD was characterized by slightly higher IQs than ASD samples and this aspect may have influenced the results. The age difference was a significant moderator in the second and third contrasts, meaning that, when the age gap was wider (i.e., when TD samples were older than both ADHD and ASD), EF impairment in ADHD and ASD became more evident. This finding is consistent with previous studies [74,75]: Van Eylen and colleagues [75] found that both age and IQ had significant effects on EF performance in a sample of children with ASD vs. typical development, with older children showing better performances than younger children. They also pointed out that this correlation between age and EF was true only considering the neuropsychological tests and not when considering BRIEF scores. This is in line with what we found when we carried out questionnaire analyses: the age difference was not a significant moderator for any of the contrasts considered. Regarding the moderation analyses of questionnaires, IQ difference resulted as a significant moderator in TD vs. ADHD contrast. Within informant measures, children and adolescents with lower IQs had greater EF problems in everyday life. In this contrast, the sex distribution difference was also a significant moderator so that when TD had a bigger number of females, differences in EF ratings diminished. An interesting finding emerged concerning the kind of rater involved in evaluating the children/adolescents. When the raters were the children, the strength of the effect size in the comparisons of clinical groups with the TD group was equal (*d*_TD vs. ADHD_ = 0.76 and *d*_TD vs. ASD_ = 0.768), meaning that when the rater was a child/adolescent with ADHD, their own perception of their EF difficulties in everyday life was basically the same perception experienced by a subject with ASD. Indeed, the effect size associated with the comparison of the two clinical groups was low (*d* = 0.017). Different impressions, however, emerged when the rater was an external respondent: compared to the TD group, both ADHD and ASD were associated with the greatest executive impairment when the rater was a parent, although ADHD was characterized by a bigger effect size (*d* = 2.191) than ASD (*d* = 1.831): a possible explanation of these different ratings is that parents of children and adolescents with ASD may tend to give them less autonomy than parents of children with ADHD and this could lead to a minor perception of their executive difficulties.

Considering the third comparison between the two clinical groups, the biggest effect size was not associated with the parents’ report anymore: in this case, the greatest executive impairment was experienced by the teachers, who rated the EF-related behaviors of children with ADHD higher than those of children with ASD. This result could be explained by the fact that EF problems may especially affect and invalidate a school environment where the children/adolescents spend much of their time. For this reason, EF difficulties can become particularly evident to teachers, whose work can be more affected by problematic behaviors such as a child’s inability to sit quietly at their own desk, a child’s inability to plan their school activities properly, difficulty in time management, a child’s inability to do their homework within time limits, etc., features characterizing more children with ADHD than with ASD [3,76,77] (especially when individuals with ASD have an IQ ≥ 70, like in our samples).

## 5. Conclusions

The aim of the present meta-analysis was to shed more light on the difference in EF profiles of two neurodevelopmental conditions, ASD and ADHD, which the literature previously shows as typically characterized by EF impairments. This paper, examining 36 studies, offers different analyses, comparing ADHD vs. TD, ASD vs. TD, ADHD vs. ASD, ADHD vs. ADHD+ASD, and ASD vs. ADHD+ASD. Moreover, for the first three analyses, the difference between the types of instruments used (neuropsychological tests vs. questionnaires) was taken into account. Both ADHD and ASD display EF impairments when compared to the TD group, particularly when assessed through questionnaires. The comparison between ADHD vs. ASD points out that these two disorders show equivalent EF impairments, with no differences within the single EF domains, only when neuropsychological tests were considered. Instead, when assessed through questionnaires, ADHD appears to be characterized by a greater executive difficulty than ASD: EF-questionnaires seem to be more predictive of everyday life EF impairments. The comparison between the double diagnosis group (ADHD+ASD) and the clinical groups pointed out that the former has a more similar EF profile to ADHD-alone and that it shows poorer performance than ASD-alone.

## 6. Limits of the Study and Future Directions

Some limitations must be taken into account: firstly, we did not separate analyses for the three clinical manifestations of ADHD (ADHD-I; ADHD-H; and ADHD-C), as we considered just one global group in which we included the different clinical manifestations; this may have contributed to creating heterogeneity.

Another limitation can be found in the intrinsic multidimensionality and complexity of EF and in the large variety of instruments (especially neuropsychological tests) used to capture the construct. We are aware that there are different types of inhibition, comprising response inhibition and interference control [78] but we did not separate the analyses and we treated them as a unitary construct that converged within the “inhibition” component. The same can be said for the instruments used: we based our instruments’ classification on the scientific literature [2,79] in order to clarify which instrument assesses which component but we are aware of the “task impurity problem”, a phenomenon in which one task assesses various EF components beyond the one it aims to evaluate (for example, WCST is sometimes considered a general EF task, requiring WM too and not only cognitive flexibility; Stroop test has been used as a measure of WM other than of inhibition [14]). All these factors can explain the high heterogeneity we found across all contrast, a limitation that the reader should be aware of in order to interpret our results more cautiously.

A third limitation regards the absence of investigation of the socioeconomic status (SES) as a possible moderator; the scientific literature has proved that a relationship between SES and EF exists so that lower SES is generally associated with poorer EF performances [33]. However, the impossibility of exploring this potential moderator was caused by the difficulty of retrieving detailed information about the SES of the samples, especially in older studies, and by the heterogeneity of measures used to assess and represent the SES (parents’ educational level, income, etc.). Further analyses are needed to explore the potential role of this moderator and the influences that it may have on the results. The same can be said about the description of cultural and linguistic factors that we have not considered and about medication use: six studies gave no information at all and we are aware that this may have affected the interpretation of results.

A final limit is that we found fewer studies assessing EF through questionnaires than those that assessed them through neuropsychological tests. Hopefully, research will address this topic in the future, allowing it to shed light on the EF profile of clinical groups with an exhaustive evaluation that includes both performance-based and report-based measures. In addition to this purpose, future studies should focus on the comparison between clinical groups through questionnaires with a factorial structure (such as BRIEF) in order to understand where these EF differences lie, reflecting higher impairments in the emotional, behavioral, or cognitive aspects of the EF.

Our study suggests some clinical implications: first, EF, as indexed by performance-based measures and questionnaires, looks somewhat different for children with ADHD vs. those with ASD and consequently indirect measures (questionnaires, rating scales, etc.) seem able to catch differential features of the EF between ASD and ADHD, aspects that could be more blurred and less evident with the assessment of EF through performance-based tasks. The difficulties caused by an EF impairment may in fact be more obvious in real-world settings, directly investigated using questionnaires, whereas experimental settings (the ones in which neuropsychological tests are used) are characterized by a rigid structure, with clearer rules and less frequent unexpected events, aspects that could lead to a finer EF performance and consequently to the lack of differentiation of the performance of two neurodevelopmental disorders both characterized by an EF weakness.

A clinical implication considering these results is that, in order to obtain an assessment as comprehensive, detailed, and grounded in daily-life difficulties as possible, both measures (i.e., neuropsychological tests and questionnaires) should be used during the evaluation. EF should not be used or considered as a clinical marker for either of the two disorders, since both ADHD and ASD show EF impairments. However, a report of daily life EF difficulties, as captured by questionnaires, may be more able to describe the functioning of people with ADHD and consequently rating scales could help clinicians to structure the most appropriate intervention. This means that, if one were to choose, for instance lacking time or economic resources, he should opt to use questionnaires instead of neuropsychological tests; the latter could be less useful to promptly provide an extensive assessment of daily-life difficulties. Also, this evidence reminds clinicians to include questionnaires in their everyday practice, since they provide a quick and helpful understanding of children’s EF.

## Figures and Tables

**Figure 1 children-11-00473-f001:**
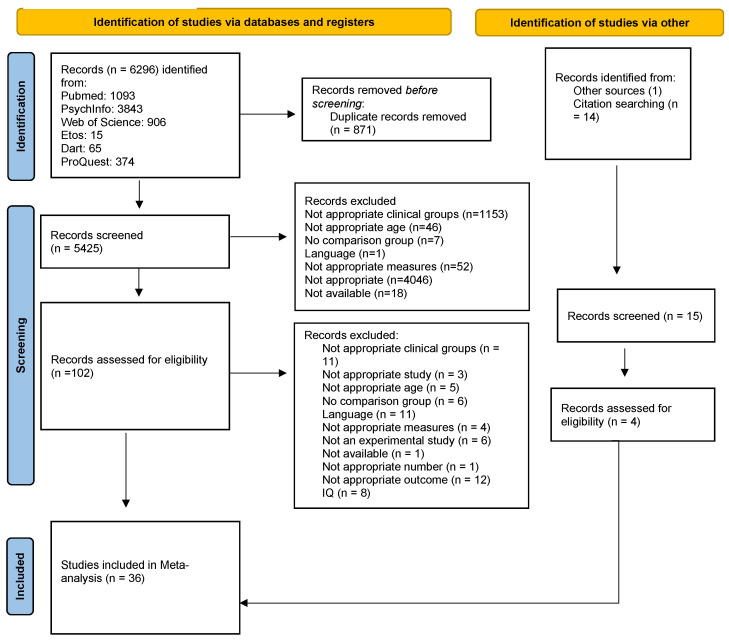
This figure shows the flow chart presenting the method of literature search and exclusion criteria.

**Table 1 children-11-00473-t001:** Descriptive Statistics of the samples included in the meta-analysis.

	ADHD		ASD		TD		ADHD+ASD		
	(*n* = 1612)		(*n* = 1049)		(*n* = 1801)		(*n* = 298)		
*Variable*	*Mean*	*SD*	*Mean*	*SD*	*Mean*	*SD*	*Mean*	*SD*	*Group. Sign*
Age	10.5	1.262	10.9	1.481	10.6	1.323	10.3	0.747	n.s.
IQ	103	5.74	102.5	6.00	110	5.33	104	3.29	TD > ADHD, ASD, ADHD+ASD
Females	278 (17%)		84 (8%)		468 (26%)		32 (11%)		n.s.

*Note.* ADHD = Attention deficit/hyperactivity disorder; ASD = Autism spectrum disorder; TD = Typical developing; ADHD+ASD = Comorbidity Group.

**Table 2 children-11-00473-t002:** Results of standardized EFs differences between ADHD, ASD, and TD globally and divided on the type of measure.

	ADHD vs. ASD	ADHD vs. TD	ASD vs. TD
*d* Global EFs	−0.03	0.73 *	0.64 *
Type of Measure	*Q* = 4.796 *p* = 0.0029	*Q* = 53.658 *p* < 0.001	*Q* = 35.984 *p* < 0.001
*d EFs Test*	0.02	0.54 *	0.51 *
*d EFs Questionnaires*	−0.34 *	1.76 **	1.34 **

*Note. Italic text* indicates the level of the categorical variable “type of measure”; *d* global EFs = standardized difference between groups in global EFs performance; *d* EFs Test = standardized difference between groups in a neuropsychological test; *d* EFs questionnaires = standardized difference between groups in questionnaires; *Q* = heterogeneity statistic; ***** *p* < 0.05; ****** *p* < 0.001.

**Table 3 children-11-00473-t003:** Standardized differences in executive functioning between ADHD, ASD, and TD in the studies included in the meta-analysis neuropsychological test.

		Effect Size (*d*)	
Study	ADHD vs. ASD	TD vs. ADHD	TD vs. ASD
Corbett, 2009 [36]	0.74	0.15	0.84
Boxhoorn, 2018 [37]	−0.72	3.17	1.84
Li, 2017 [38]	−0.09	0.42	0.23
Geurts, 2004 [39]	0.39	0.25	0.63
Goldberg, 2005 [21]	0.19	0.24	0.44
Gomarus, 2009 [40]	0.38	−0.08	0.26
Hutchison, 2016 [41]	0.27	0.55	0.73
Hwang-Gu, 2019 [42]	−0.30	0.65	0.33
Johnson, 2007 [43]	−0.72	1.34	0.67
Mahone, 2006 [22]	−0.35	0.96	0.54
Matsuura, 2014 [44]	−0.04	0.55	0.45
Nydén, 1999 [45]	−0.31	1.22	0.70
Ozonoff, 1999 [46]	0.43	0.24	0.63
Karalunas, 2018 [47]	−0.06	0.46	0.39
Pitzianti, 2016 [48]	−0.16	1.81	1.43
Salunkhe, 2021 [49]	−0.09	0.33	0.22
Samyn, 2015 [50]	0.02	0.06	0.09
Samyn, 2017 [51]	0.09	0.14	0.24
Semrud-Clikeman, 2010a [52]	0.74	0.16	0.87
Semrud-Clikeman, 2010b [53]	0.41	0.326	0.71
Sinzig, 2008a [54]	−0.34	0.46	0.06
Sinzig, 2008b [55]	−0.26	0.26	−0.02
Tsuchiya, 2005 [56]	0.05	1.29	1.26
Tye, 2014 [57]	−0.25	0.28	0.03
Unterrainer, 2016 [58]	−0.35	0.20	−0.12
Unterrainer, 2020 [59]	−0.16	0.42	0.22
Verté, 2006 [60]	0.15	0.62	0.72
Wang, 2018 [61]	−0.04	0.67	0.70
Xiao, 2012 [62]	0.23	0.58	0.60
Yasumura, 2014 [63]	−0.15	0.19	0.05
Azadi Sohi, 2012 [64]	−0.20	0.37	0.19
Manteris, 2011 [65]	0.61	−0.22	0.40
Shahabuddin, 2015 [66]	−0.25	1.81	1.95

## Data Availability

We have uploaded the dataset used in the meta-analysis to an online repository (https://osf.io/6wugn/?view_only=46270f6337fa46b984b891bfc4c99138) accessed on 5 December 2021.

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
