# Peer review of "Comparing Executive Functions in Children and Adolescents with Autism and ADHD—A Systematic Review and Meta-Analysis"

_children, 2024, doi:10.3390/children11040473_

Round 1

Reviewer 1 Report

Comments and Suggestions for Authors

This systematic review and meta-analysis explored, comparatively, the executive functions in children and adolescents with ADHD and autism. The efforts of the Authors to conduct such a review that included scientific databases and gray literature is praiseworthy. Please refer to the following observations:

Line 35- please avoid using „EF dysfunction”, maybe replace this by „EF impairment”;

Lines 58-59- please insert a reference for DSM-5;

Lines 70-83- this sentence is far too long, consider splitting it for better comprehension;

Lines 154- „neuropsychological tests”;

Within the „Methodology” section, a table containing the PICO algorithm for selecting studies would be useful;

Line 205- the preregistration site is not available in the manuscript;

Lines 327-328- the quality assessment of the cross-sectional studies is possible, and there are tools for this process, e.g., AXIS, Newcastle-Ottawa Scale, etc.

Author Response

Reviewer 1

This systematic review and meta-analysis explored, comparatively, the executive functions in children and adolescents with ADHD and autism. The efforts of the Authors to conduct such a review that included scientific databases and gray literature is praiseworthy. Please refer to the following observations:

  • Line 35- please avoid using “EF dysfunction”, maybe replace this by “EF impairment”;

We replaced EF dysfunction” using “EF impairment.

  • Lines 58-59- please insert a reference for DSM-5;

We inserted the reference for DSM-5.

  • Lines 70-83- this sentence is far too long, consider splitting it for better comprehension;

We split the sentence in four parts. The text has been highlighted in yellow.

  • Lines 154- “neuropsychological tests”;

We corrected the word “test” in “tests”.

  • Within the „Methodology” section, a table containing the PICO algorithm for selecting studies would be useful;

The PICO algorithm was presented as PRISMA flow chart in figure 1.

  • Line 205- the preregistration site is not available in the manuscript;

We decided not to write the registration site to guarantee the blindness of the authorship, but you can find it here: https://www.crd.york.ac.uk/prospero/display_record.php?RecordID=232689

  • Lines 327-328- the quality assessment of the cross-sectional studies is possible, and there are tools for this process, e.g., AXIS, Newcastle-Ottawa Scale, etc.

The quality assessment of the cross-sectional studies is possible indeed, however we decided not to examine the quality of the publication but we focused our attention on the publication bias.

Reviewer 2 Report

Comments and Suggestions for Authors

Thank you for the opportunity to read Comparing Executive Functioning in children with autism and ADHD. Overall, the manuscript is well written with sound analyses of the data. I have a few suggestions that I believe would help strengthen the manuscript. The research question in the abstract and on page 3 starting line 143 are not clearly written. It isn't necessary in a manuscript to include the hypotheses with the research questions since the results are being reported in the same paper. The main question is written as an intervention study rather than a study for a meta analytic paper. The overall question would be more appropriate if written to determine a more accurate estimate of the differences regarding EF between individuals with ADHD, ASD and TD based upon two measures. The phrasing in section 4 line 431 is a better representation of the purpose of the study than the other research questions earlier in the paper. 

I strongly suggest the authors reflect on the use of deficit language (worse/better, damaged) and use less negative, judgmental language to describe outcomes. 

Specificity is also recommended regarding who completed the questionnaires as that could be a factor that influences the data. It is also not completely clear why this study is needed given the information included in section 2.6. 

A major issue is the inclusion of moderating factors in the discussion but these factors were not addressed in the results. There is a great deal of detail in the discussion without any data to support it. 

Another factor that influences the likelihood of publishing this study is the numerous limitations. While it is appreciated that the authors recognize the issues, the limitations themselves limit the usability of data to make the claims within the study. 

Minor suggestions include: a) change the use of "subjects" line 60 to individuals; b) add a semicolon in your list of inclusion criteria after each item in the list; c) include what years the studies were from in your search criteria; address the XXXs in line 204 as it is unclear what they refer to and why that information is represented as XXX; d) clarify what is meant by line 247- does that information refer to the what happened during data extraction or is it a list of the studies?; e) clarify what section 2.5 refers to and why those methods are sufficient to address quality; f) the location and reference to Table 3 is confusing in 3.6 as there isn't a column in the table that addresses ADHD+ASD; g) page 12 is one long paragraph- break it up into sections for readability; and h) there are very few implications for the study- why is this study needed if claims and directions for practice can't be made about EF and individuals with ADHD and/or ASD?

Attention to these suggestions could increase the opportunity for publication. 

Comments on the Quality of English Language

Minor issues as noted in the previous section are noted with suggestions for improvement. 

Author Response

Reviewer 2

Thank you for the opportunity to read Comparing Executive Functioning in children with autism and ADHD. Overall, the manuscript is well written with sound analyses of the data. I have a few suggestions that I believe would help strengthen the manuscript.

  • The research question in the abstract and on page 3 starting line 143 are not clearly written. It isn't necessary in a manuscript to include the hypotheses with the research questions since the results are being reported in the same paper. The main question is written as an intervention study rather than a study for a meta analytic paper. The overall question would be more appropriate if written to determine a more accurate estimate of the differences regarding EF between individuals with ADHD, ASD and TD based upon two measures. The phrasing in section 4 line 431 is a better representation of the purpose of the study than the other research questions earlier in the paper. 

We rephrased the research question in the abstract and in page 3 starting line 145.

  • I strongly suggest the authors reflect on the use of deficit language (worse/better, damaged) and use less negative, judgmental language to describe outcomes. 

We used a less negative language in several sentences of the paper.

  • Specificity is also recommended regarding who completed the questionnaires as that could be a factor that influences the data.

We addressed this issue in the results and discussion.

  • It is also not completely clear why this study is needed given the information included in section 2.6. 

According to our opinion this study is necessary following the four research questions reported on page 3.

  • A major issue is the inclusion of moderating factors in the discussion but these factors were not addressed in the results. There is a great deal of detail in the discussion without any data to support it. 

A new paragraph about the moderation analyses has been added in the results section.

  • Another factor that influences the likelihood of publishing this study is the numerous limitations. While it is appreciated that the authors recognize the issues, the limitations themselves limit the usability of data to make the claims within the study. 

We decided to reduce the limitation to the core ones.

Minor suggestions include:

  1. change the use of "subjects" line 60 to individuals;

We changed the word “subjects”.

  1. add a semicolon in your list of inclusion criteria after each item in the list;

We added semicolons in our list.

  1. include what years the studies were from in your search criteria;

We included the years range in the criteria.

  1. address the XXXs in line 204 as it is unclear what they refer to and why that information is represented as XXX;

We decided not to write the registration site to guarantee the blindness of the authorship, but you can find it here: https://www.crd.york.ac.uk/prospero/display_record.php?RecordID=232689

  1. clarify what is meant by line 247- does that information refer to the what happened during data extraction or is it a list of the studies?

We clarified that the information refers to the database.

  1. clarify what section 2.5 refers to and why those methods are sufficient to address quality;

We decided not to examine the quality of the publication but to focus on publication bias

  1. the location and reference to Table 3 is confusing in 3.6 as there isn't a column in the table that addresses ADHD+ASD;

We changed the position of Table 3.

  1. page 12 is one long paragraph- break it up into sections for readability;

We divided page 12 up in sections.

  1. there are very few implications for the study- why is this study needed if claims and directions for practice can't be made about EF and individuals with ADHD and/or ASD?

We further stressed the paper implications.

Round 2

Reviewer 2 Report

Comments and Suggestions for Authors

Thank you for the response to previous feedback regarding the manuscript. I have additional suggestions for the authors to consider. The research questions are more clear in the updated manuscript. The language is more appropriate and the inclusion of information regarding moderating factors is helpful.

In line 14, it seems like a word or two is missing - could change "depending upon" the type of measure.

The content in the supplementary files is critical. The inclusion of this content in a meta analysis is essential. It is highly recommended that the supplemental content, particularly the tables, be included in the manuscript itself. I know that would extend the manuscript substantially. The included study descriptions could be accessed via the database with the link provided in line 238 and authors listed in Table 3, but the results tables are very helpful. That may be an issue for the editor as I found them very helpful as a reader. 

Shouldn't lines 352, 356, 364 be "lower"? Line 366 wording "finer" is confusing. I had to re-read the included sentence starting line 706 several times so clarity within that point should be addressed.

Comments on the Quality of English Language

Suggestions for clarity were included in the previous comments.

Author Response

Thank you for the response to previous feedback regarding the manuscript. I have additional suggestions for the authors to consider. The research questions are more clear in the updated manuscript. The language is more appropriate and the inclusion of information regarding moderating factors is helpful.

  • In line 14, it seems like a word or two is missing - could change "depending upon" the type of measure.

We corrected the sentence.

  • The content in the supplementary files is critical. The inclusion of this content in a meta analysis is essential. It is highly recommended that the supplemental content, particularly the tables, be included in the manuscript itself. I know that would extend the manuscript substantially. The included study descriptions could be accessed via the database with the link provided in line 238 and authors listed in Table 3, but the results tables are very helpful. That may be an issue for the editor as I found them very helpful as a reader. 

The link of the supplementary material is reported at line 712, however whe have asked to the editor if it is more appropriate to include the tables in the manuscript.

  • Shouldn't lines 352, 356, 364 be "lower"?

Yes, they should. We corrected them.

  • Line 366 wording "finer" is confusing.

We substituted “finer” with “higher”.

  • I had to re-read the included sentence starting line 706 several times so clarity within that point should be addressed.

We modified this sentence in order to clarify it.